

# Low-energy effective theory and anomalous Hall effect in monolayer WTe$_2$

**Snehasish Nandy and D. A. Pesin**

Department of Physics, University of Virginia, Charlottesville, VA 22904 USA

## Abstract

We develop a symmetry-based low-energy theory for monolayer WTe$_2$ in its 1T′ phase, which includes eight bands (four orbitals, two spins). This model reduces to the conventional four-band spin-degenerate Dirac model near the Dirac points of the material. We show that measurements of the spin susceptibility, and of the magnitude and time dependence of the anomalous Hall conductivity induced by injected or equilibrium spin polarization can be used to determine the magnitude and form of the spin-orbit coupling Hamiltonian, as well as the dimensionless tilt of the Dirac bands.

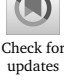

## 1 Introduction

The two dimensional topological insulator has been a prime topic of interest in recent years due to its intriguing properties as well as being a potential candidate for technological applications.

After initial theoretical proposals, quantum-well heterostructures based on three-dimensional semiconductors (e.g., HgTe/CdTe and InAs/GaSb quantum wells) were at the center of the experimental attention, see Refs. [1,2] for review.

Recently, the focus has shifted to truly two-dimensional materials, in particular, transitional metal dichalcogenides. Among these, monolayer WTe$_2$ in its 1T$'$ phase with a large bulk band gap ($\sim$ 0.055 eV) has been theoretically proposed as a candidate material for quantum spin Hall insulator [3–6]. Soon after the theoretical prediction, several experiments have been done, which strongly supported the presence of helical edge channels, as well as quantized electronic conductance of $\frac{2e^2}{h}$ over a large range of temperatures (up to 100 K) [7–13]. In addition to the quantum spin Hall state, the distorted 1T$'$ WTe$_2$ shows interesting phenomena associated with various types of interactions, such as gate-induced superconductivity [14,15], exciton condensation [16], nonlinear edge magnetotransport [17], and possibly charge density wave [18,19].

Typically, various attempts of theoretical description of the aforementioned variety of interesting phenomena, (for instance, see Refs. [20–23]), rely on some low-energy models, presumed capable of capturing the relevant physics. However, the first-principles understanding of 1T$'$ WTe$_2$, which usually forms the basis for construction of low-energy models, has been steadily evolving. There are now several proposals regarding the exact pattern of band inversion in this material, and the closely related question of the symmetry and the orbital nature of the states at the $\Gamma$-point [3, 4, 24–28]. It can be said that most of theoretical works that employ a type of low-energy theory derive that from the symmetry analysis based on Ref. [3], in which the conduction and valence bands closest to the Fermi level were assumed to have opposite parities.

In this paper, we develop effective low-energy $\mathbf{k}\cdot\mathbf{p}$ model Hamiltonians for monolayer WTe$_2$ in its 1T$'$ phase, and discuss predictions for spin dynamics and anomalous charge transport that follow from these models. We start out with an eight-band model, using the symmetry analysis based on recent first-principle calculations [4, 25–28]. The key observation made in those works is that the orbitals that give rise to the conduction and valence bands closest to the Fermi level have the same parity. This fact affects the form of the spin-orbit coupling, the leading part of which is momentum-independent near the $\Gamma$-point. Subsequently, we reduce the eight-band model to a four-band one valid near the Dirac points, and discuss the implications of this model for the spin dynamics and anomalous charge transport in monolayer WTe$_2$.

The rest of the paper is organized as follows. In Section 2, we develop low-energy $\mathbf{k} \cdot \mathbf{p}$ model Hamiltonians of monolayer WTe$_2$ using symmetry analysis. Section 3 is devoted to physical implications of the models developed in Section 2. Finally, in Section 4 we discuss the obtained results.

## 2 Low-energy models near the $\Gamma$-point

The monolayer WTe$_2$, which belongs to the space group P2$_1$/m, is the only material among TMDCs to show the topologically non-trivial 1T$'$ structure as its stable ground state and gives rise to quantum Hall insulating phase [3–13, 17]. In the absence of SOC, the bulk states of monolayer WTe$_2$ show a two-dimensional Dirac semimetal phase due to presence of two tilted gapless Dirac cones near the $\Gamma$ point, which are protected by the nonsymmorphic glide-mirror symmetry [25, 27, 28]. When spin-orbit coupling is included, small gaps close to the Dirac points appear. Our immediate goal is to start with a $\mathbf{k} \cdot \mathbf{p}$-model near the $\Gamma$-point that involves the eight bands (two spin projections included) near the Fermi level, and derive a reduced four-band model valid near the two Dirac points. In doing so, we generalize the six-band analysis of Ref. [20].

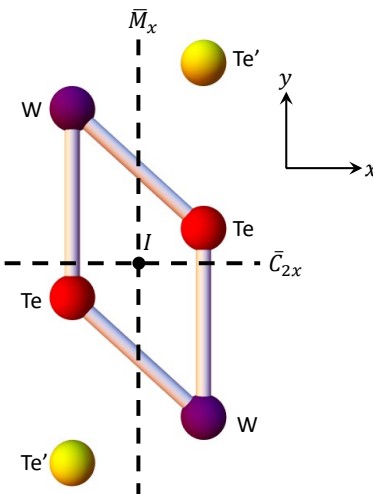

Figure 1: (Color online) The unit cell of monolayer WTe$_2$. $d$-orbitals centered at purple tungsten, W, sites, and $p$-orbitals centered at red tellurium, Te, cites, give rise to the low-energy bands near the Fermi level. The orbitals of the yellow Te$'$ sites do not participate in the low-energy physics. Also shown are the glide plane $\bar{M}_x$, and the screw axis $\bar{C}_{2x}$ that run through the inversion center $I$.

## 2.1 Eight-band model

The 1T$'$ structure of monolayer WTe$_2$ belongs to C$_{2h}^2$ point symmetry group. Its unit cell is show in Fig. 1. It has the following symmetries [25, 27, 28]: (i) lattice translations t($\hat{x}$) and t($\hat{y}$), (ii) two-fold screw symmetry around the $x$ axis: $\bar{C}_{2x}$=t($\hat{x}/2$)$C_{2x}$ which is the product of a two-fold rotation $C_{2x}$ and a translation t($\hat{x}/2$) by half a lattice vector, (iii) a glide mirror symmetry: $\bar{M}_x$=t($\hat{x}/2$)$M_x$, which is the product of $M_x$–reflection with respect to the $yz$-plane, and a translation, the same as for screw symmetry, and (iv) time-reversal symmetry. Since, $M_x$ operates as $(x,y,z) \rightarrow (-x,y,z)$ and $C_{2x}$ operates as $(x,y,z) \rightarrow (x,-y,-z)$, the combined operation of these two operators i.e. $\bar{C}_{2x}\bar{M}_x$ gives rise to spatial inversion $\mathcal{I}$ that sends $\mathbf{r} \rightarrow -\mathbf{r}$. The four irreducible representations of C$_{2h}^2$, and the corresponding character table is shown in Table. 1.

There are four nondegenerate (apart from the Kramers degeneracy) states at the $\Gamma$-point, which are close to the Fermi level. We denote them as $\{\psi_1, \psi_2, \psi_3, \psi_4\}$. The lowest conduction and highest valence bands, which form the Dirac points, derive from the two middle orbitals, $\psi_2$ and $\psi_3$, respectively. According to recent DFT analysis [4,9,25,28], these orbitals transform according to $\Gamma_1$ and $\Gamma_2$ irreducible representations, respectively, as shown in Table. 1. They have the same parity, but opposite mirror eigenvalues. The highest conduction band, $\psi_1$, and lowest valence band, $\psi_4$, transform according to $\Gamma_3$ and $\Gamma_4$ representations, respectively (see Fig. 4 of the Appendix).

A general eight-band $\mathbf{k} \cdot \mathbf{p}$ model Hamiltonian can be written as

$$\hat{H} = \sum_{\alpha,\beta} |\psi_\alpha\rangle\langle\psi_\beta|\mathcal{H}^{\alpha\beta}(\mathcal{K}), \tag{1}$$

where $\mathcal{H}^{\alpha\beta}(\mathcal{K})$ is a $2 \times 2$ (in spin space) matrix element connecting states $\psi_\alpha$ and $\psi_\beta$, and $\mathcal{K}$ denotes a tensor operator formed by combinations of wave vectors.

Since $|\psi_\alpha\rangle\langle\psi_\alpha|$ is even under all symmetry operations mentioned in Table. 1 (i.e., $\Gamma_i \otimes \Gamma_i = \Gamma_1$), the blocks $\mathcal{H}^{\alpha\alpha}(\mathcal{K})$, therefore, must be composed of $\mathcal{K}$ operators that are also even under those symmetry operations. For each off-diagonal blocks, we can make the following observations:

Table 1: Character Table of $C_{2h}$ point group. $\Gamma_i$'s are the irreducible representations of the group.

| $C_{2h}$ | I | $\mathcal{I}$ | $M_x$ | $C_{2x}$ |
|---|---|---|---|---|
| $\Gamma_1$ | 1 | 1 | 1 | 1 |
| $\Gamma_2$ | 1 | 1 | -1 | -1 |
| $\Gamma_3$ | 1 | -1 | -1 | 1 |
| $\Gamma_4$ | 1 | -1 | 1 | -1 |

i) $H^{12}(\mathcal{K})$ and $H^{34}(\mathcal{K})$: Both $|\psi_1\rangle\langle\psi_2|$ and $|\psi_3\rangle\langle\psi_4|$ are even under $C_{2x}$ and odd under both $M_x$ and $\mathcal{I}$ and therefore transform according to $\Gamma_3$ representation.

ii) $H^{13}(\mathcal{K})$ and $H^{24}(\mathcal{K})$: Both $|\psi_1\rangle\langle\psi_3|$ and $|\psi_2\rangle\langle\psi_4|$ are even under $M_x$ and odd under both $\mathcal{I}$ and $C_{2x}$, therefore transform according to $\Gamma_4$.

iii) $H^{14}(\mathcal{K})$ and $H^{23}(\mathcal{K})$: Both $|\psi_1\rangle\langle\psi_4|$ and $|\psi_2\rangle\langle\psi_3|$ are odd under both $M_x$ and $C_{2x}$ and even under $\mathcal{I}$ and therefore transform according to $\Gamma_2$ representation.

So, these off-diagonal blocks must be composed of $\mathcal{K}$ operators that also transform under their corresponding $\Gamma$ representation. Now, we know $M_x : (k_x, k_y) \rightarrow (-k_x, k_y)$, $(\sigma_1, \sigma_2, \sigma_3) \rightarrow (\sigma_1, -\sigma_2, -\sigma_3)$; $C_{2x} : (k_x, k_y) \rightarrow (k_x, -k_y)$, $(\sigma_1, \sigma_2, \sigma_3) \rightarrow (\sigma_1, -\sigma_2, -\sigma_3)$; $\mathcal{I} : (k_x, k_y) \rightarrow (-k_x, -k_y)$, $(\sigma_1, \sigma_2, \sigma_3) \rightarrow (\sigma_1, \sigma_2, \sigma_3)$; and TRS sends $\boldsymbol{\sigma} \rightarrow -\boldsymbol{\sigma}$ and $\boldsymbol{k} \rightarrow -\boldsymbol{k}$. Following the above operations the terms that can appear in the off-diagonal blocks of this $\mathbf{k} \cdot \mathbf{p}$ Hamiltonian are given in Table. 2.

Table 2: Terms (upto second-order in momentum) that can appear for eight-band $\mathbf{k} \cdot \mathbf{p}$ Hamiltonian blocks.

| $H^{\alpha\alpha}(\boldsymbol{k})$ | $H^{12}(\boldsymbol{k})$, $H^{34}(\boldsymbol{k})$ | $H^{13}(\boldsymbol{k})$, $H^{24}(\boldsymbol{k})$ | $H^{14}(\boldsymbol{k})$, $H^{23}(\boldsymbol{k})$ |
|---|---|---|---|
| $\sigma_0$, $k_x^2\sigma_0$, $k_y^2\sigma_0$, $i\sigma_1$, $ik_x^2\sigma_1$, $ik_y^2\sigma_1$ | $ik_x\sigma_0$, $k_x\sigma_1$, $k_y\sigma_2$, $k_y\sigma_3$ | $ik_y\sigma_0$, $k_y\sigma_1$, $k_x\sigma_2$, $k_x\sigma_3$ | $k_xk_y\sigma_0$, $i\sigma_2$, $i\sigma_3$, $ik_xk_y\sigma_1$, $ik_x^2\sigma_2$, $ik_x^2\sigma_3$, $ik_y^2\sigma_2$, $ik_y^2\sigma_3$ |

Since the $\mathbf{k} \cdot \mathbf{p}$ Hamiltonian is Hermitian, the diagonal block of the Hamiltonian ($H^{\alpha\alpha}(\boldsymbol{k})$) should also be Hermitian and therefore, $ik_x^2\sigma_1$, $ik_y^2\sigma_1$, $i\sigma_1$ will not appear in the Hamiltonian. Using $(\psi_{1\uparrow}, \psi_{1\downarrow}, \psi_{2\uparrow}, \psi_{2\downarrow}, \psi_{3\uparrow}, \psi_{3\downarrow}, \psi_{4\uparrow}, \psi_{4\downarrow})$ as the basis, the full eight-band $\mathbf{k} \cdot \mathbf{p}$ model Hamiltonian (up to $O(\boldsymbol{k})$ in off-diagonal terms) in the absence of the spin-orbit coupling can be written as

$$
\mathcal{H}_0^{k\cdot p} = \begin{pmatrix}
\epsilon_1 & 0 & iv_1k_x & 0 & iv_2k_y & 0 & d_1k_xk_y & 0 \\
0 & \epsilon_1 & 0 & iv_1k_x & 0 & iv_2k_y & 0 & d_1k_xk_y \\
-iv_1k_x & 0 & \epsilon_2 & 0 & d_2k_xk_y & 0 & iv_3k_y & 0 \\
0 & -iv_1k_x & 0 & \epsilon_2 & 0 & d_2k_xk_y & 0 & iv_3k_y \\
-iv_2k_y & 0 & d_2k_xk_y & 0 & \epsilon_3 & 0 & iv_4k_x & 0 \\
0 & -iv_2k_y & 0 & d_2k_xk_y & 0 & \epsilon_3 & 0 & iv_4k_x \\
d_1k_xk_y & 0 & -iv_3k_y & 0 & -iv_4k_x & 0 & \epsilon_4 & 0 \\
0 & d_1k_xk_y & 0 & -iv_3k_y & 0 & -iv_4k_x & 0 & \epsilon_4
\end{pmatrix}, \quad (2)
$$

Table 3: Value of parameters for the low-energy model given in Eq. (2). These values are obtained after fitting this low-energy model to the tight-binding model (see Eq. (33)).

| Parameter | Value | Unit | Parameter | Value | Unit |
|---|---|---|---|---|---|
| $c_{01}$ | 1.48 | eV | $c_{11}$ | -1.33 | eV·Å$^2$ |
| $c_{02}$ | 1.12 | eV | $c_{12}$ | -0.15 | eV·Å$^2$ |
| $c_{03}$ | -0.12 | eV | $c_{13}$ | -1.13 | eV·Å$^2$ |
| $c_{04}$ | -0.96 | eV | $c_{14}$ | 0.11 | eV·Å$^2$ |
| $c_{31}$ | 0.18 | eV·Å$^4$ | $c_{21}$ | -0.02 | eV·Å$^2$ |
| $c_{32}$ | 0.08 | eV·Å$^4$ | $c_{22}$ | -1.50 | eV·Å$^2$ |
| $c_{33}$ | 0.18 | eV·Å$^4$ | $c_{23}$ | 0.02 | eV·Å$^2$ |
| $c_{34}$ | 0.07 | eV·Å$^4$ | $c_{24}$ | 1.50 | eV·Å$^2$ |
| $v_1$ | -1.02 | eV·Å | $d_1$ | -0.24 | eV·Å$^2$ |
| $v_2$ | 0.18 | eV·Å | $d_2$ | 0.24 | eV·Å$^2$ |
| $v_3$ | -1.77 | eV·Å | $\lambda$ | 0.04 | eV·Å$^3$ |
| $v_4$ | 1.02 | eV·Å | $V'$ | 0.04 | eV |

where $\epsilon_i = c_{0i} + c_{1i}k_x^2 + c_{2i}k_y^2 + c_{3i}k_x^4$. By fitting this model to the tight-binding one derived in Ref. [25], we can get the values of the coefficients $c_i$'s, $d_i$'s and $v_i$'s. The values of the parameters are given in Table. 3. It is important to note that although the energy dispersion obtained from the low-energy model (see Eq. (2)) matches well with the tight-binding Hamiltonian, there is a little mismatch (a few meV) between the dispersions close to the Dirac nodes. This discrepancy appears because the Dirac nodes are quite far from the Γ-point in the Brillouin zone, whereas our low-energy model has been written near the Γ-point. A near-perfect match of the $\boldsymbol{k} \cdot \boldsymbol{p}$-model dispersion with the tight-binding one can be obtained if one includes additional $\lambda k_x^3$ terms in the interband couplings in Eq. (2). The value of $\lambda$ is also given in Table. 3. Since the tight-binding model cannot not reproduce exactly the energy dispersion obtained from the DFT calculations anyway [27, 28], we ignore such terms from now on. Their inclusion would not improve the accuracy of quantitative results obtained from the tight-binding model.

We now turn to the discussion of the spin-orbit coupling. We ignore possible Rashba spin-orbit coupling terms, which may arise in this system by breaking inversion symmetry due to a substrate. According to Ref. [27], the leading spin-orbit coupling term within the eight-band manifold near the Fermi level stems from the spin-flip hops along the W-Te bonds that lie in the symmetry plane, see Fig. 1. Within the $\mathbf{k} \cdot \mathbf{p}$ model, this corresponds to $k$-independent spin-orbit coupling term written as

$$\mathcal{H}_{SO}^{k \cdot p} =
\begin{pmatrix}
0 & 0 & 0 & 0 & 0 & 0 & iV' & V \\
0 & 0 & 0 & 0 & 0 & 0 & -V & -iV' \\
0 & 0 & 0 & 0 & iV' & V & 0 & 0 \\
0 & 0 & 0 & 0 & -V & -iV' & 0 & 0 \\
0 & 0 & -iV' & -V & 0 & 0 & 0 & 0 \\
0 & 0 & V & iV' & 0 & 0 & 0 & 0 \\
-iV' & -V & 0 & 0 & 0 & 0 & 0 & 0 \\
V & iV' & 0 & 0 & 0 & 0 & 0 & 0
\end{pmatrix}, \tag{3}$$

Table 4: Terms that can appear for four-band $\mathbf{k} \cdot \mathbf{p}$ Hamiltonian blocks.

| $H^{\alpha\alpha}(\boldsymbol{k})$ | $H^{12}(\boldsymbol{k})$ |
|---|---|
| $\sigma_0,\ \ i\sigma_1,\ \ k_x^2\sigma_0,\ \ k_y^2\sigma_0,$ $ik_x^2\sigma_1, ik_y^2\sigma_1$ | $k_x k_y\sigma_0,\quad i\sigma_2,\quad i\sigma_3,$ $ik_x k_y\sigma_1,\ ik_x^2\sigma_2,\ ik_x^2\sigma_3,$ $ik_y^2\sigma_2, ik_y^2\sigma_3$ |

where parameters $V$ and $V'$ describe the spin-orbit coupling strength. The value of $\sqrt{V^2 + V'^2}$, which we take to be 40 meV in this work, determines the size of the gap around the Fermi energy, and their relative magnitude defines the orientation of the conserved spin projection for the low-energy model, see Section 2.2 for further details.

Corrections to the $k$-independent spin-orbit coupling (3) stem from other spin-flip hopping paths, and appear to be small in first-principles calculations [28]. Furthermore, as will be shown in Section 2.2, the leading $k$-independent spin-orbit coupling gives rise to the conservation of the spin projection on a particular axis near the $\Gamma$-point. This is consistent with the sample, edge-orientation, and gate-voltage independent spin quantization axis on edges of a sample, as experimentally observed [17]. This provides us with empirical evidence that the spin-orbit coupling (3) indeed describes the states close to the Fermi level in monolayer WTe$_2$. The sub-leading spin-orbit coupling terms, which are quadratic in the quasimomentum and would break conservation of any spin projection away from the $\Gamma$-point, are not important.

Taken together, the Hamiltonians (2) and (3) define the eight-band $\boldsymbol{k} \cdot \boldsymbol{p}$-model for monolayer WTe$_2$.

## 2.2 Four-band model near Dirac points

Since the two Dirac nodes appears in the bulk due to middle two bands, we now want to develop a four-band (including spin) low-energy $\mathbf{k} \cdot \mathbf{p}$ model Hamiltonian near the $\Gamma$ point. As mentioned above, both of these bands have same parity eigenvalues and opposite mirror eigenvalues, the screw eigenvalues are also opposite for these bands. Aside from the spin degrees of freedom, each of these orbitals at the $\Gamma$ point is nondegenerate and transforms according to one of the irreducible representations of the Table. 1.

Using basis $(\psi_{c\uparrow}, \psi_{c\downarrow}, \psi_{v\uparrow}, \psi_{v\downarrow})$, the generalized low-energy four-band $\mathbf{k} \cdot \mathbf{p}$ model Hamiltonian (up to $O(\boldsymbol{k}^2)$ in off-diagonal terms) near $\Gamma$ point in the absence of SOC for ML WTe$_2$ can be written as

$$
\mathcal{H}_0^{k\cdot p} = \begin{pmatrix} \epsilon_c & 0 & dk_x k_y & 0 \\ 0 & \epsilon_c & 0 & dk_x k_y \\ dk_x k_y & 0 & \epsilon_v & 0 \\ 0 & dk_x k_y & 0 & \epsilon_v \end{pmatrix}, \tag{4}
$$

where $\epsilon_\alpha = a_{0\alpha} + a_{1\alpha}k_x^2 + a_{2\alpha}k_y^2 + a_{3\alpha}k_x^4$ with $\alpha = c, v$. Here, the coefficients $a_\alpha$'s and $d$ can be expressed in terms of $c_i$'s, $d_i$'s and $v_i$'s of the eight-band low-energy model. The relation between the coefficients of these two models can be obtained using the method as described in Ref. [20]. Using the similar analysis described in ref. [20], the effective four-band $\mathbf{k} \cdot \mathbf{p}$ model Hamiltonian extracted from the eight-band $\mathbf{k} \cdot \mathbf{p}$ model Hamiltonian given in Eq. (2) can be written as

$$
\mathcal{H}_0^{k\cdot p} = S^{-1/2}(h_q - u h_d^{-1} u^\dagger) S^{-1/2}, \tag{5}
$$

where

$$
h_d = \begin{pmatrix} \epsilon_2 & 0 & d_2 k_x k_y & 0 \\ 0 & \epsilon_2 & 0 & d_2 k_x k_y \\ d_2 k_x k_y & 0 & \epsilon_3 & 0 \\ 0 & d_2 k_x k_y & 0 & \epsilon_3 \end{pmatrix},
$$

$$
h_q = \begin{pmatrix} \epsilon_1 & 0 & d_1 k_x k_y & 0 \\ 0 & \epsilon_1 & 0 & d_1 k_x k_y \\ d_1 k_x k_y & 0 & \epsilon_4 & 0 \\ 0 & d_1 k_x k_y & 0 & \epsilon_4 \end{pmatrix},
$$

$$
w = \begin{pmatrix} -iv_1 k_x & 0 & iv_3 k_y & 0 \\ 0 & -iv_1 k_x & 0 & iv_3 k_y \\ -iv_2 k_y & 0 & iv_4 k_x & 0 \\ 0 & -iv_2 k_y & 0 & iv_4 k_x \end{pmatrix},
$$

$$
S = 1 + w h_d^{-2} w^\dagger. \tag{6}
$$

We find that the expressions of $a_\alpha$'s and $d$ in terms of $c_i$'s, $d_i$'s and $v_i$'s are cumbersome. Therefore, one can simply get the values of $a_\alpha$'s and $d$ by fitting this model to the eight-band model. For example, we find that $d = 0.56$ eV·Å$^2$ for the four-band Hamiltonian given in Eq. (4) at the Dirac point $\boldsymbol{k} \approx (0.84, 0)^{-1}$.

Now, we will discuss the SOC in this model. Using the same basis of the Eq. (4), the leading order SOC term in the $\mathbf{k} \cdot \mathbf{p}$ model can be written as

$$
\mathcal{H}_{SO}^{k \cdot p} = V_1 \tau_2 \sigma_2 + V_1' \tau_2 \sigma_3 \equiv \tau_2 \boldsymbol{V_s} \cdot \boldsymbol{\sigma}, \tag{7}
$$

where $V_1$ and $V_1'$ are the spin-orbit coupling strength. Here, $V_s = (0, V_1, V_1')$ can be thought of as a vector lying in the mirror plane making an angle of $\tan^{-1}(\frac{V_1}{V_1'})$ with the $z$ axis [17].

In the absence of the spin-orbit coupling, the lowest conduction and highest valence bands are degenerate at two points located at the $x$-axis perpendicular to the mirror plane. We label these two Dirac points as $K_\xi$, with the valley index $\xi = \pm 1$. The effective four-band Hamiltonian describing electronic states near each of such Dirac points is obtained by adding the spin-orbital coupling Hamiltonian (7) to Hamiltonian (4), and expanding in $\boldsymbol{k}$ near a Dirac point. It follows from Eqs. (4) and (7) that close to either of $K_\pm$, the low-energy Hamiltonian, written in the basis of the conduction and valence bands making up the Dirac cone, is given by

$$
H_\xi = \xi \sigma_0 (-u k_x \tau_0 + v_x k_x \tau_3 + v_y k_y \tau_1) + \Delta_{so} (\boldsymbol{\sigma} \cdot \boldsymbol{d_s}) \tau_2, \tag{8}
$$

where Pauli matrices $\tau$ act in the space of the conduction and valence bands, $v_x$ and $v_y$ are the Fermi velocity along $x$ and $y$ directions respectively, $\Delta_{so}$ is the spin-orbit coupling strength, and $u$ describes band tilting. The unit vector $\boldsymbol{d_s}$ points along $\boldsymbol{V_s}$ of Eq. (7), while the spin-orbit coupling parameter $\Delta_{so}$ is given by the magnitude of $\boldsymbol{V_s}$. Each band that the Hamiltonian describes is doubly degenerate at every $\boldsymbol{k}$-point, as appropriate for a system with both time-reversal and inversion symmetries. The corresponding tilted Dirac dispersions are shown in the inset of Fig. 4 in the Appendix.

There are several notable features of Hamiltonian (8). It is clear from the form of Hamiltonian (8) that the projection of spin onto the axis defined by $\boldsymbol{d_s}$ is a good quantum number for Hamiltonian (8), since effectively only a single ($\boldsymbol{\sigma}$) Pauli matrix enters the Hamiltonian, so one can replace $\boldsymbol{\sigma} \cdot \boldsymbol{d_s}$ with its eigenvalue, $\boldsymbol{\sigma} \cdot \boldsymbol{d_s} \to \eta = \pm 1$ while diagonalizing (8). Furthermore, the dispersions of the conduction, $c$, and valence, $v$, bands near the Fermi level are given by $E_k^{c,v} = -\xi u k_x \pm \sqrt{(v_x^2 k_x^2 + v_y^2 k_y^2 + \Delta_{so}^2)}$. Because of the band tilt, the band extrema are shifted from $K_\xi$. If we denote $u/v_x \equiv \beta > 0$, the conduction band minima are

located at $K_\xi + (\xi k_{min}, 0)$, such that $v_x k_{min}/\Delta_{so} = \beta/\sqrt{1-\beta^2}$. The full indirect gap is $E_g = 2\Delta_{so}\sqrt{1-\beta^2}$. To determine the parameters of the low-energy model, we used those reported for the tight-binding model of Ref. [25], but reduced the strength of the spin-orbit coupling to match the experimental value of the gap, which is around 55 meV [9]. As a result, we obtained $\Delta_{so} = 40$ meV, $v_x = 6.44 \times 10^5$ m/s, $v_y = 3.65 \times 10^5$ m/s and $\beta = 0.72$.

We note that previous works all arrived at similar forms of four-band Hamiltonians near Dirac points, see, for instance, Refs. [3,20–22]. This is despite the fact that the assumed symmetry of the conduction and valence band states was different in those studies. For instance, in Refs. [3,21,22], the mirror symmetries of the conduction and valence bands were taken to be the same, while their parities are opposite. As a result, the leading spin-orbit coupling is linear in momentum component perpendicular to the mirror plane, and evaluates to a constant near a Dirac point. This leads to the same type of the spin-orbit coupling near a Dirac point as in Eq. (8). In Ref. [20], both the mirror symmetries, and parities of the conduction and valence band states at the $\Gamma$-point are assumed to be opposite. This leads to a spin-orbit coupling linear in the momentum component in the mirror plane, while the gap at the Dirac points on the axis perpendicular to the mirror plane is opened even without spin-orbit coupling. Even in the case of Ref. [20], a unitary transformation can bring the four-band Hamiltonian studied there to a form used in Ref. [3]. In view of this analysis of existing works, we would like to emphasize that despite the uniformity of four-band models near the Dirac point, they look quite different away from them. This fact can be important for studies of many-body physics, like exciton condensation [16], which is sensitive to the wave functions of the bands both near the Dirac points, and the $\Gamma$-point, or optical phenomena near the $\Gamma$-point. In such cases, it is probably best to use the eight-band model of Eqs. (2) and (3) of the present work.

The predictions of the low-energy model near a Dirac point, Eq. (8), for the spin susceptibility, spin splitting, and the anomalous Hall effect will be studied in the rest of this paper. The bulk spin transport for this model, and its optical properties were studied in Refs. [22] and [29], respectively.

# 3 Physical consequences of the low-energy model

Below we will be interested in the behavior of electronic spins in monolayer WTe$_2$ in a magnetic field. We will restrict ourselves to energies close to the bottom of the conduction bands, located at momenta near the Dirac points. These minima are much more pronounced than the valence band maxima due to a considerable band tilting near Dirac points.

## 3.1 Effective Zeeman coupling

The coupling of electronic spin to a magnetic field stems from the associated magnetic moment. For electrons in a crystal, this magnetic moment can be divided into an intrinsic part related to the electronic spin, and an orbital part related to the rotation of electronic wave packets moving throughout the crystal [30]. We will first describe the orbital contribution to the effective $g$-factor.

In general, one expects all three Cartesian components of the carrier orbital magnetic moment to have nonzero values in bulk monolayer WTe$_2$. However, the in-plane ones stem from the relative atomic displacements in the $z$-direction, which are of the order of a few Å. On the contrary, the $z$-component of the magnetic moment comes from the in-plane spread of the electronic wavepackets, and near a Dirac point the scale of this spread is given by the effective Compton wavelength, $\hbar v_D/\Delta_{so} \sim 100$ Å, where $v_D \sim 5 \times 10^5$m/s is the typical value of the Dirac speed, which is anisotropic in monolayer WTe$_2$, as mentioned above. There-

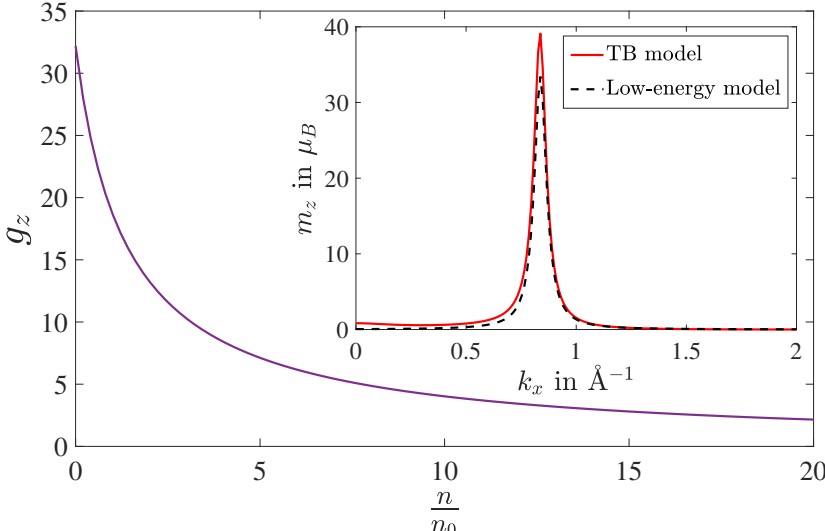

Figure 2: (Color online) Depicts the effective orbital $g$-factor for out-of-plane magnetic field, $g_z$ as a function of $\frac{n}{n_0}$. Inset shows the magnitude of the orbital magnetic moment (in units of $\mu_B$) as a function of $k_x$ for the lowest conduction band using tight-binding model (solid line) and the Hamiltonian given in Eq. (8) (dashed line). Here we have taken $\Delta_{so} = 40$ meV, $v_x = -6.44 \times 10^5$ m/s and $v_y = 3.65 \times 10^5$ m/s.

fore, the effects associated with the in-plane orbital magnetic moments are suppressed by two orders of magnitude with respect to the out-of-plane one, and we will neglect them below. The expression for the $z$-component of the orbital moment of the Dirac electron described by Hamiltonian (8) has the standard form [30]:

$$m_z(\boldsymbol{k}) = \frac{e v_x v_y \Delta_{so}}{2\epsilon_{\boldsymbol{k}}^2}(\boldsymbol{\sigma} \cdot \boldsymbol{d_s}), \tag{9}$$

where $\epsilon_{\boldsymbol{k}} = \sqrt{v_x^2 k_x^2 + v_y^2 k_y^2 + \Delta_{so}^2}$. The lack of dependence on the valley index, $\xi$, in Eq. (9) reflects the presence of the inversion symmetry.

In the inset of Fig. 2, we compare the orbital magnetic moment of the conduction band electrons calculated from the low-energy Hamiltonian (8), and the eight-band tight-binding model of Ref. [25]. It is clear that the magnetic moment calculated from the low-energy theory agrees well with the results of a more general model. It is also apparent that the orbital magnetic moment depends strongly on the momentum. For instance, for $\boldsymbol{\sigma} \cdot \boldsymbol{d_s} = +1$, and at the Dirac point, it is given by $m_z(\boldsymbol{k} = 0) = e v_x v_y / 2\Delta_{so} \approx 33\mu_B$, where $\mu_B < 0$ is the Bohr magneton; for the same spin projection, but at the bottom of the conduction band, the orbital moment becomes $m_z(k_{min}) = e v_x v_y (1 - \beta^2)/2\Delta_{so} \approx 16\mu_B$.

The aforementioned strong momentum dependence of the orbital magnetic moment signals that the effective $g$-factor for out-of-plane magnetic fields depends strongly on the doping level. At low doping, it comes predominantly from the orbital moment of electrons, and can be expressed through its value averaged over the Fermi contour:

$$\overline{m}_z = \frac{1}{v(E_F)} \int (d\boldsymbol{k}) m_z(\boldsymbol{k})\delta(E_{\boldsymbol{k}} - E_F) = e v_x v_y \frac{\Delta_{so}(1 - \beta^2)^{3/2}}{E_F \sqrt{E_F^2 + \beta^2 \Delta_{so}^2}} \boldsymbol{\sigma} \cdot \boldsymbol{d_s}. \tag{10}$$

In the above expression, we introduced the density of states per spin per valley:

$$\nu(E_F) = \frac{E_F}{2\pi v_x v_y (1-\beta^2)^{3/2}}\,. \tag{11}$$

It can be useful in practice to give the values of the effective $g$-factor as a function of density, rather than of the Fermi energy. The dependence of the total density on the Fermi level is given by

$$n(E_F) = N_s N_v \frac{E_F^2 - \Delta_{so}^2(1-\beta^2)}{4\pi v_x v_y (1-\beta^2)^{3/2}}\,, \tag{12}$$

where $N_s = N_v = 2$ are the spin and valley degeneracies, respectively. Eq. (12) shows that the characteristic scale for the density is given by

$$n_0 = N_s N_v \frac{\Delta_{so}^2}{4\pi v_x v_y (1-\beta^2)^{1/2}} \approx 8\cdot 10^{11}\text{cm}^{-2}\,. \tag{13}$$

We introduce the effective orbital $g$-factor for out-of-plane magnetic field, $g_z$, using the average spin splitting on the Fermi surface, given by $2|\overline{m}_z|B_z \equiv \mu_B g_z B_z$. Eqs. (10), (12), and (13) then allow to plot $g_z$ as a function of $n/n_0$, which is done in Fig. 2. The plot suggests that for out-of-plane magnetic fields, the spin-Zeeman effects become comparable to the orbital ones for $n/n_0 \sim 10$.

The spin part of the Zeeman coupling is dominant for in-plane magnetic fields. To describe it, we will assume that the atomic $g$-factors of the monolayer WTe$_2$ constituents are all equal to 2. That is not necessarily the case in general in view of the atomic spin-orbit coupling. However, in the case of WTe$_2$, the atomic spin-orbit coupling is quenched in the orbitals comprising the low-energy manifold, so one expects that the deviation from the free-space value are small in the ratio of the spin-orbit coupling strength and crystal field splitting for each atom, which should be small for extended $5p$ and $5d$ orbitals.

With the bare $g$-factor being equal to 2, the spin-Zeeman term in the 4-band Hamiltonian near a Dirac point reads

$$H_Z^{spin} = -\mu_B(\boldsymbol{\sigma} \cdot \boldsymbol{B})\tau_0\,, \tag{14}$$

where $\boldsymbol{B}$ is external magnetic field. We are interested in the matrix elements of this Hamiltonian in the space spanned by the two spin-degenerate conduction band close to the Fermi level. It is clear from Hamiltonian (8) that the wave functions of these states can be written as $|s, \chi_{s\xi}\rangle$, where $s$ is the eigenvalue of the spin projection onto $\boldsymbol{d}_s$, such that $\boldsymbol{\sigma}\cdot\boldsymbol{d}_s|s, \chi_{s\xi}\rangle = s|s, \chi_{s\xi}\rangle$, with $s = \pm$, $\xi$ is the valley index, and $\chi_{s\xi}$ labels the corresponding states in the orbital space. Explicitly, one choice for these states is given by

$$|\chi_{s\xi}\rangle = \sqrt{\frac{1}{2} + \frac{\xi v_x k_x}{2\epsilon_k}} \left(1, is\frac{\epsilon_k - \xi v_x k_x}{\Delta + is\xi v_y k_y}\right)^T\,. \tag{15}$$

The overlaps of these orbital states strongly affect the matrix elements of the spin-Zeeman term. For instance, at a Dirac point one has $k_x = k_y = 0$, and thus $\langle\chi_{s\xi}|\chi_{s'\xi}\rangle = \delta_{ss'}$, hence a magnetic field oriented in any direction can at most shift the two spin states in energy (as long as it is not orthogonal to $\boldsymbol{d}_s$), but cannot flip the spin.

There is a large degree of arbitrariness to the matrix elements of the spin-Zeeman Hamiltonian between the low-energy states because of arbitrary relative $\boldsymbol{k}$-dependent phase conventions that can be chosen for them. Different choices of the relative phase amount to a unitary transformation performed on the Hamiltonian. Therefore, these matrix elements are

not observable, and do not define any physical $g$-tensor, and they do not directly define the effective Zeeman field that determines spin precession. What is observable is the spin splitting generated by these matrix elements (the spectrum, of the Hamiltonian), and the associated spin susceptibility, as well as the effective fields that enter the Heisenberg equation of motion for the spin operator.

The spin susceptibility for in-plane Zeeman fields was calculated numerically in Ref. [21] for the case of the conduction and valence band states having the opposite parity, but the same mirror eigenvalues. Here we provide the corresponding analytic expressions for the present case.

Below we will specialize to $K_+$ valley, setting $|\chi_{s\xi}\rangle \to |\chi_\pm\rangle$ for $s = \pm$. Because of the inversion symmetry of the system, the expressions for physical observables obtained below are exactly the same for $K_-$ valley. First, we define the matrix elements of the spin polarization in the basis of the two spin-degenerate conduction band states,

$$\Sigma^a_{s,s'} = \langle s, \chi_s | \sigma_a \tau_0 | s', \chi_{s'} \rangle, \tag{16}$$

where the placement of the Cartesian upper index $a$ on $\Sigma^a$ is done only for notational convenience. Note that $\boldsymbol{\Sigma} \cdot \boldsymbol{d}_s$ acts like $\sigma_z$ in the basis of the degenerate conduction band states. Using matrices $\Sigma$, the effective Zeeman Hamiltonian, which includes both the spin and orbital magnetic moment contributions, is written as

$$H_Z^{tot} = -\mu_B \boldsymbol{\Sigma} \cdot \boldsymbol{B} - m_z(\boldsymbol{k}) B_z \boldsymbol{\Sigma} \cdot \boldsymbol{d}_s. \tag{17}$$

It is apparent that the orbital magnetic moment leads to an effective Zeeman field directed along $\boldsymbol{d}_s$, and whose magnitude is given by $m_z(\boldsymbol{k}) B_z / \mu_B$. For out of plane fields, this effective field dominates over the spin-related counterpart at low doping levels, due to the large value of $m_z$ at the conduction band edge. For in-plane fields it vanishes.

## 3.2 In-plane spin susceptibility

We start with a discussion of the in-plane spin susceptibility. It was suggested in Refs. [14,15], and numerically explored in Ref. [21], that the reduction of the normal-state in-plane spin susceptibility may be a reason for the enhancement of the in-plane critical field in gate-induced superconductivity in monolayer WTe$_2$ beyond the paramagnetic limit. Below we present an analytic calculation of this quantity.

The expression for the in-plane spin susceptibility $\chi^{ab}$ has the standard linear-response form,

$$\chi^{ab}(E_F) = \frac{N_v}{2} \mu_B^2 \int (d\boldsymbol{k}) \text{Tr}\{\Sigma^a(\boldsymbol{k}), \Sigma^b(\boldsymbol{k})\} \delta(E_{\boldsymbol{k}} - E_F), \tag{18}$$

where $\{\dots,\dots\}$ denotes the anticommutator. Further considerations are simplified if one introduces the projectors onto the direction of $\boldsymbol{d}_s$, $P_\parallel$, and onto the plane perpendicular to $\boldsymbol{d}_s$, $P_\perp$. These projectors satisfy $P_\parallel + P_\perp = \mathbb{1}$, and are given by

$$P_\parallel^{ab} = d_s^a d_s^b; \quad P_\perp = \delta^{ab} - d_s^a d_s^b. \tag{19}$$

In terms of these projectors one easily obtains

$$\text{Tr}\{\Sigma^a(\boldsymbol{k}), \Sigma^b(\boldsymbol{k})\} = 4\left(P_\parallel^{ab} + |\langle\chi_+|\chi_-\rangle|^2 P_\perp^{ab}\right). \tag{20}$$

It is apparent that Eq. (20), and hence Eq. (18), are not sensitive to the choice of an arbitrary relative phase for the two conduction band states. The overlap that enters Eq. (20) is given by

$$|\langle\chi_+|\chi_-\rangle|^2 = \frac{\epsilon_{\boldsymbol{k}}^2 - \Delta_{so}^2}{\epsilon_{\boldsymbol{k}}^2}. \tag{21}$$

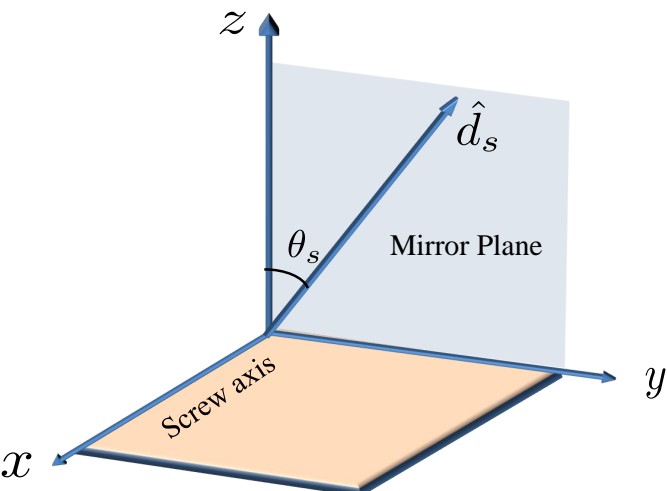

Figure 3: (Color online) Schematic diagram of the spin-orientation axis $\boldsymbol{d_s}$ which makes an angle $\theta_s$ from the $z$-axis in monolayer WTe$_2$.

Near the conduction band bottom, where $\epsilon_{\boldsymbol{k}} = \Delta_{so}/\sqrt{1-\beta^2}$, we have $|\langle \chi_+ | \chi_- \rangle|^2 \approx \beta^2$.

The spin susceptibility of Eq. (18) is essentially the Fermi-surface average of the trace of $\Sigma$-matrices given in Eq. (20). Performing the necessary integrations, which are the same as in the case of the orbital moment, one obtains

$$\chi^{ab}(E_F) = N_v \chi_0 \left[ P_{\parallel}^{ab} + \left( 1 - \frac{\Delta_{so}^2 (1-\beta^2)^{3/2}}{E_F \sqrt{E_F^2 + \beta^2 \Delta_{so}^2}} \right) P_{\perp}^{ab} \right], \tag{22}$$

where $\chi_0 = 2\mu_B^2 \nu(E_F)$.

It can be seen from Eq. (22) that for $\Delta_{so} \sim 40$ meV, and $E_F \sim 100$ meV, for which the doping density is around $10^{13}$cm$^{-2}$, that the difference between components of $\chi^{ab}$ and $\chi_0$ is at most a few percent. It does not seem plausible that the difference between $\chi$ and $\chi_0$ can explain the aforementioned increase.

It is somewhat curious that in the low-doping limit, where the Fermi energy approaches the bottom of the conduction band, $E_F \rightarrow \Delta_{so}\sqrt{1-\beta^2}$, the $\chi^{xx}$ susceptibility is reduced by a factor of $\beta^2 \approx 0.5$ as compared to the naive value $2\mu_B^2 \nu(E_F)$ for $g = 2$. The reason $\chi^{xx}$ vanishes for $\beta = 0$ is, again, the fact that the orbital parts of the two degenerate states at the bottom of the conduction band are then orthogonal to each other, and there is no response to the magnetic field to linear order in $B$.

If the tilt, $\beta$, is known, then the orientation of $\boldsymbol{d_s} \equiv (0, \sin\theta_s, \cos\theta_s)$ can be inferred (see Fig. 3) from the ratio of the diagonal in-plane susceptibilities in the $E_F \rightarrow \Delta_{so}\sqrt{1-\beta^2}$ limit:

$$\sin^2 \theta_s = \frac{\beta^2}{1-\beta^2} \left( \frac{\chi^{yy}}{\chi^{xx}} - 1 \right). \tag{23}$$

Therefore, we conclude that the in-plane spin susceptibility in the low-doping limit can be directly linked to the properties of the low-energy Hamiltonian: the tilt, and the orientation of the conserved spin direction.

## 3.3 Anomalous Hall effect in and out of equilibrium

In the presence of a magnetic field, one expects WTe$_2$ to show Hall effect. For out-of-plane field direction, the Hall effect is dominated by the usual semiclassical Drude value, and has

little to do with the underlying wave functions of the low-energy Dirac model. However, due to the low symmetry of the sample, even an in-plane magnetic field, which does not produce substantial orbital effects (but still has some due to the monolayer extension in the $z$-direction, see above) can induce anomalous Hall effect. A similar effect induced by the planar magnetic field has also been proposed in two-dimensional hole gas (p-type semiconductors) [31] and trigonal crystals with sizable spin-orbit coupling [32]. The effect is the strongest close to the bottom of the conduction band, where the Berry curvature is the largest. Since the spin projection onto $d_S$ is a good quantum number in the low-energy theory, the Berry curvature for the degenerate conduction bands can be calculated for each of the spin projections, the result being the standard expression for the Berry curvature of a two-level system, described by Hamiltonian (8):

$$\Omega_z(\boldsymbol{k}) = -\frac{v_x v_y \Delta_{so}}{2\epsilon_{\boldsymbol{k}}^3}(\boldsymbol{\Sigma} \cdot \boldsymbol{d_s}). \tag{24}$$

This expression does not depend on the valley index $\xi$, so it is suppressed. In the low-energy limit, $\epsilon_{\boldsymbol{k}} \to \Delta_{so}/\sqrt{1-\beta^2}$, it is obvious that the Berry curvature is directly proportional to the spin projection onto $d_s$. Therefore, the momentum-space integral of its expectation value, which determines the anomalous Hall conductivity [30], can be expressed through the spin susceptibility of Eq. (22). Before writing down the expression for the anomalous Hall conductivity for energies near the conduction band bottom, we note that one must include the side-jump contribution, since it is parametrically identical to the intrinsic one, described by the Berry curvature of Eq. (24). For a Dirac model, and at low energies the side-jump contribution is twice as big in magnitude, and opposite in sign as compared to the intrinsic one [33]. Effectively, its inclusion just changes the sign of the intrinsic contribution. Keeping this sign change in mind, using Eqs. (24) and (22), and the fact that $d_s^a P_\parallel^{ab} = d_s^b$, $d_s^a P_\perp^{ab} = 0$, one can write the anomalous Hall conductivity as

$$\sigma_{xy}^{AHE} = N_v \frac{e^2}{2\pi} \frac{|\mu_B| \boldsymbol{B} \cdot \boldsymbol{d_s}}{\Delta_{so}}. \tag{25}$$

We emphasize that the magnetic field enters the expression for the Hall conductivity via the Zeeman splitting of the bands, which removes the degeneracy between the two bands with opposite values of the Berry curvature. It is apparent from Eq. (25) that if the orientation of $\boldsymbol{d_s}$ is known, measuring the anomalous Hall conductivity for in-plane magnetic field oriented in the mirror plane provides a direct way to measure the strength of the spin-orbit coupling in the low-energy model. The orientation of $\boldsymbol{d_s}$ can be determined as in Ref. [17].

The anomalous Hall conductivity (25) was obtained under the conditions of equilibrium, when the spin polarization of a sample is described by susceptibility (22). However, experiments on spin polarization injection into monolayer dichalcogenides are also commonplace. It then follows from Eq. (24) that transient spin dynamics manifests itself in the Hall conductivity of monolayer WTe$_2$. That is, the Hall signal provides a direct window into the spin dynamics.

To understand what information can be extracted from the time dependence of the Hall signal driven by nonequilibrium spin polarization, one has to study the spin precession dynamics in WTe$_2$. The equations of motion for the spin polarization operators can be written in Heisenberg representation with Hamiltonian (17). For simplicity, we will assume that an in-plane magnetic field is applied, $B_z = 0$. Generalization to arbitrary field direction is trivial, and amounts to the replacement $\boldsymbol{B} \to \boldsymbol{B} + m_z B_z \boldsymbol{d_s}$, as follows from Eq. (17). We also do not consider spin relaxation here, which can be added phenomenologically, if needed. The equations of motion for the Cartesian components of the spin polarization then read

$$\frac{d\Sigma^a}{dt} = i[H_Z^{tot}, \Sigma^a], \tag{26}$$

which has to be supplemented with the expression for the commutator of $\Sigma$-matrices:

$$[\Sigma^a, \Sigma^b] = 2if^{abc}\Sigma^c. \tag{27}$$

The third-rank pseudotensor $F^{abc}$ has the following form:

$$f^{abc} = \epsilon^{abl}\left(P_\perp^{lc} + |\langle\chi_+|\chi_-\rangle|^2 P_\parallel^{lc}\right). \tag{28}$$

To proceed, we define the expectation value of the spin polarization density with respect to the non-equilibrium single-particle density matrix,

$$S = \int (d\boldsymbol{k})\langle\Sigma(\boldsymbol{k})\rangle, \tag{29}$$

and restrict ourselves to the energies close to the conduction band bottom, $|\langle\chi_+|\chi_-\rangle|^2 \approx \beta^2$. This allows us to write a closed set of equations for the spin polarization density using the equations of motion for $\Sigma$. In components, the equation of motion for $S$ looks as follows:

$$\frac{dS^a}{dt} = -2\mu_B f^{abc}B^b S^c. \tag{30}$$

It is convenient to use the decomposition $\boldsymbol{S} = S_\parallel \boldsymbol{d}_S + \boldsymbol{S}_\perp$, such that $\boldsymbol{S}_\perp \cdot \boldsymbol{d}_s = 0$, since $S_\parallel$ directly determines the Hall conductivity. Then we obtain

$$\begin{aligned}
\frac{dS_\parallel}{dt} &= -2\mu_B\beta^2 \boldsymbol{d}_s \cdot \boldsymbol{B} \times \boldsymbol{S}_\perp, \\
\frac{d\boldsymbol{S}_\perp}{dt} &= -2\mu_B\beta^2 \boldsymbol{B} \times \boldsymbol{d}_s S_\parallel - 2\mu_B \boldsymbol{B} \times \boldsymbol{S}_\perp + 2\mu_B(\boldsymbol{d}_s \cdot \boldsymbol{B} \times \boldsymbol{S}_\perp)\boldsymbol{d}_s.
\end{aligned} \tag{31}$$

These equations simplify in a great, and rather useful fashion when the $\boldsymbol{B}$-field is perpendicular to the mirror symmetry plane, $\boldsymbol{B} = (B, 0, 0)$. For such fields, the last two terms in the second of Eqs. (31) cancel out, and the projection of $\boldsymbol{S}_\perp$ onto $\boldsymbol{B}$ does not change in time. Finally, $S_\parallel$, and the projection of $\boldsymbol{S}_\perp$ onto $\boldsymbol{d}_s \times \boldsymbol{B}$, which we will denote simply as $S_\perp$, evolve according to

$$\begin{aligned}
\frac{dS_\parallel}{dt} &= -2\mu_B B S_\perp, \\
\frac{dS_\perp}{dt} &= 2\mu_B\beta^2 B S_\parallel.
\end{aligned} \tag{32}$$

The spin polarization rotates around the direction of $\boldsymbol{B}$ with the modified Larmor frequency, given by $\omega_L = 2\beta|\mu_B|B$, reduced by a factor of $\beta$ from its free-space value. Clearly, the same oscillation pattern is inherited by the anomalous Hall conductivity. Its oscillation frequency is thus a direct measure of the tilt parameter $\beta$.

## 4 Conclusions

In this work, we have developed the $\boldsymbol{k} \cdot \boldsymbol{p}$ model that describes the eight (four orbitals times two spins) bands near the Fermi level of monolayer WTe$_2$, Eqs. (2) and (3). We further reduced it to a four-band model valid near the Dirac points, Eq. (8). The eight-band model should be useful for studies of exciton condensation, which involves both states near the $\Gamma$-point and Dirac points [16], and optical phenomena near the $\Gamma$-point. The four-band model is convenient to

consider Fermi-surface type of phenomena, of which we considered the anomalous transport, magnetic susceptibility for in-plane fields, and the effective $g$-factor for out-of-plane fields.

The choice of the above list of quantities to consider was motivated by the fact that they provide insight into the parameters and form of the low-energy Hamiltonian that describes monolayer WTe$_2$. We have shown that most of the parameters describing this Hamiltonian can be obtained from measuring its magnetic susceptibility, as well as the anomalous Hall effect driven by equilibrium and non-equilibrium - injected - spin polarizations. Such measurements can help to determine the orientation of the conserved spin projection, the dimensionless tilt of band dispersion, and the strength of the spin-orbit coupling, see Eqs. (23), (25), and (32). Perhaps to measure such quantities in a single experiment can be considered difficult. However, the direction of $d_s$, inferred from edge transport, was shown to be very robust from sample to sample in Ref. [17], so the angle $\theta_s$ can be considered known. Further information about this quantity can be inferred from spin transport measurements proposed in Ref. [22]. Then purely electrical measurements of the Hall conductivity provide full information about the dimensionless band tilt, and the strength of the spin-orbit coupling. According to Eq. (25), the value of the Hall conductivity in units of the conductance quantum is set by the ratio of the Zeeman energy and the spin-orbit strength. For $B \sim 1$T, it is of order of $10^{-3}$, and is measurable in experiment. If the strength of the spin-orbit coupling and tilt are known, measurements of the effective $g$-factor for out-of-plane fields yield information about the geometric mean of the Dirac velocities, see Eq. (10), and the discussion in the preceding paragraph.

Overall, the results of this work should be useful for experimental characterization of monolayer WTe$_2$ samples, as well as for future studies of many-body and optical phenomena in monolayer WTe$_2$.

## Acknowledgements

We are grateful to David Cobden and Wenjin Zhao for extensive discussions and collaboration, which led to this work. We would also like to acknowledge useful discussions of their published work with Anton Akhmerov, Alexander Lau, Lukas Muehler, Ronny Thomale, Likun Shi, Justin Song, Jose Hugo García, and Stephan Roche. This work was supported by the National Science Foundation Grant No. DMR-1853048.

## A  Tight-binding Hamiltonian of monolayer WTe$_2$

In the absence of atomic spin-orbit coupling, including the spin $s = \uparrow, \downarrow$, sublattice $c = A, B$ and orbital $l = d, p$ degrees of freedom, the minimal eight-band tight-binding Hamiltonian in momentum space for monolayer WTe$_2$ is given by [25, 27]

$$\mathcal{H}_0 = \sigma_0 \otimes \begin{pmatrix} \epsilon_d & 0 & \tilde{t}_d g_k e^{ik_y} & \tilde{t}_{01} f_k \\ 0 & \epsilon_p & -\tilde{t}_{02} f_k & \tilde{t}_p g_k \\ \tilde{t}_d g_k^* e^{-ik_y} & -\tilde{t}_{02} f_k^* & \epsilon_d & 0 \\ \tilde{t}_{01} f_k^* & \tilde{t}_p g_k^* & 0 & \epsilon_d \end{pmatrix}, \tag{33}$$

where $\epsilon_l = \mu_l + 2t_l \cos k_x + 2t_l' \cos 2k_x$ for $l = p, d$, $g_k = 1 + e^{-ik_x}$, $f_k = 1 - e^{-ik_x}$, $\tilde{t}_l = t_l^{AB} e^{-i\boldsymbol{k} \cdot (\boldsymbol{r}_{B,l} - \boldsymbol{r}_{A,l})}$ with $l = p, d$, $\tilde{t}_{01} = t_0^{AB} e^{-i\boldsymbol{k} \cdot (\boldsymbol{r}_{B,p} - \boldsymbol{r}_{A,d})}$ and $\tilde{t}_{02} = t_0^{AB} e^{-i\boldsymbol{k} \cdot (\boldsymbol{r}_{B,d} - \boldsymbol{r}_{A,p})}$. The numerical values of the parameters are given in Ref. [27]. In the basis of the above Hamiltonian, the different symmetry operators can be represented as $\mathcal{I} = \sigma_0 \rho_1 \tau_3$, $T = i\sigma_2 \rho_0 \tau_0 K_1$

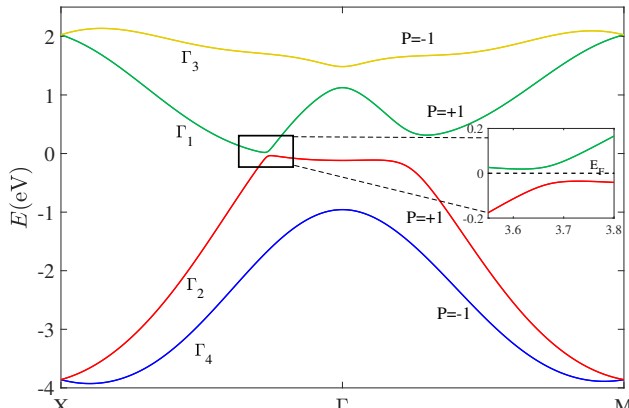

Figure 4: (Color online) Energy dispersion of the bulk bands of the monolayer $WTe_2$ along the X-$\Gamma$-M $k$ path. The middle two bands have the same parity eigenvalue $P = 1$. This indicates that the band inversion occurs between the first conduction and second valence band. We have taken $V_0' = 40$ meV.

with $K_1$ is the complex conjugation operator. For the nontranslational component of mirror and screw symmetry operator, we have $M_x = i\sigma_1\rho_0\tau_3$ and $C_{2x} = -i\sigma_1\rho_1\tau_0$ respectively. Here, the Pauli matrices $\sigma_i$, $\rho_i$ and $\tau_i$ ($i = 0, 1, 2, 3$) act on the spin, sublattice and orbital degree of freedom respectively. Each band is doubly degenerate due to presence of both TRS and inversion symmetry (IS). Using the same basis of the Eq. (33), the intrinsic SOC term with lowest order in **k** for the tight-binding model, which satisfies all the four symmetries mentioned above, can be obtained as

$$\mathcal{H}_{SO} = V_0\sigma_2\rho_3\tau_2 + V_0'\sigma_3\rho_3\tau_2, \tag{34}$$

where $V_0$ and $V_0'$ are the coefficients and two types of terms in Eq. (34) are related by spin rotation by $\pi/4$ around $\sigma_1$.

The energy dispersion of the bulk bands of the monolayer $WTe_2$ based on the tight-binding Hamiltonian along the X-$\Gamma$-M path in the momentum space is shown in Fig. 4.

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
