# Peer review of "Low-energy effective theory and anomalous Hall effect in monolayer $\mathrm{WTe}_2$"

_SciPost Physics, doi:SciPost Phys. 12, 120 (2022)_

## Round 1 · Referee Report · Anonymous · 2021-12-22

Strengths
1- Derivation from theory of invariants of a k.p low-energy model for the 1T' phase of WTe2 both in the presence and in the absence of spin-orbit coupling
2- Interesting prediction on the appearance of an anomalous Hall effect with applied in-plane magnetic fields
Weaknesses
1 - The analysis based on the theory of invariants does not bring result different from the paper by Muechler and collaborators [PRX 6, 041069 (2016)] in the absence of spin-orbit coupling.
2- The manuscript lacks some quantitative estimations. For instance, it is not immediately obvious if the anomalous Hall effect in the presence of in-plane magnetic fields can be observed in experiments.
Report
In "Low-energy effective theory and anomalous Hall effect in monolayer WTe2", the authors develop a k.p low-energy model for the 1T' phase of WTe2. The model is first derived in the absence of spin-orbit coupling, in which case the continuum k.p model reproduces the electronic properties of the tight-binding model originally developed by Muechler and coworkers in PRX 6, 041069 (2016). In a second step, spin-orbit coupling is explicitly introduced. Here the authors restrict to momentum-independent terms of spin-orbit coupling. However, a discussion of other symmetry-allowed spin-orbit coupling terms would have been beneficial. Using an additional Schrieffer-Wolf transformation, the authors then derive an effective four-band model for tilted Dirac cones that are gapped by spin-orbit coupling. In my opinion, the most interesting aspect of this manuscript is the discussion concerning the occurrence of an anomalous Hall effect triggered via Zeeman coupling by a planar magnetic field. However, I would like to point to the authors that precisely the same effect has been originally proposed by Battilomo and collaborators in Physical Review Research 3, L01006 (2021), and dubbed Anomalous planar Hall effect. Later, the same effect has been discussed by Cullen and collaborators in Physical Review Letters 126, 256601 (2021).
Requested changes
1- When deriving the spin-orbit coupling Hamiltonian in Sec.II-A, the authors should discuss if spin-orbit coupling term that are linear in k are symmetry allowed. In principle those term, if appearing, should be explicitly taken into account considering that the spin-orbit free Hamiltonian is expanded up to linear order in momentum
2- In the same subsection, the authors also mention that there is conservation of the spin-projection in their effective Hamiltonian. Is this enforced by a crystalline symmetry, as it happens for instance in graphene when preserving the Mz horizontal mirror symmetry? Or this is an accidental property due to the assumptions made on the spin-orbit coupling terms?
3 - A minor remark concerns the statement in Sec.II-B "There are several notable features of Hamiltonian. Each band that it describes is double degenerate at every k-point, as appropriate for a system with both time-reversal and inversion symmetry". I would not consider "notable" the double degeneracy of the bands in an inversion and time-reversal symmetric system.
4- I am confused by the expression for the anomalous Hall conductance due to planar magnetic field, that seems to be inversely proportional to the spin-orbit coupling strength (see Eq.25).The Berry curvature indeed grows proportionally with the SOC. Could the authors clarify the scaling and what happens to the transverse conductance in the spin-orbit free limit?
5- It would be also beneficial to estimate the size of the Hall conductance for moderate magnetic fields and estimate if the effect can be in principle observed in experiments.
Author: Snehasish Nandy on 2022-02-17 [id 2214]
(in reply to Report 1 on 2021-12-22)
We thank the referee for their report and helpful comments and suggestions. We also thank the referee for bringing these useful references (Physical Review Letters 126, 256601 (2021) and Physical Review Research 3, L012006 (2021) ) to our attention. We have now cited these references (Refs.~30 and 31) in the revised version of the manuscript. (First Paragraph of Section-IIIC, Page 8).
Please find our response to the questions raised by the referee in the attached file.
Author: Snehasish Nandy on 2022-02-17 [id 2215]
(in reply to Report 2 on 2022-02-02)We are grateful to the Referee for their report and positive assessment of our work. Please find our response to the comments from the Referee in the attached file.
Attachment:
Response_to_Report_2.pdf

---

## Round 1 · Referee Report · Anonymous · 2022-2-2

Strengths
1- It provides a comprehensive discussion of models for WTe2 monolayer. The discussion includes a description of differences of previous models in the literature which will be a useful starting point for other researchers interested in this system.
2- It makes an effort to connect detail parameters of the model with various experimental probes. This effort might help experimentalist in interpretation of results and it may also help or provide ideas for other compounds.
Weaknesses
1- The presentation (good overall) might be improved in some parts.
Report
The authors present a low-energy model for monolayer WTe2 in the inversion symmetric phase. They provide insightful discussions into the possible variants that can describe the low-energy electronic structure of this system and also presents certain phenomena that could experimentally reveal information about specific parameters of such models.
I find the paper well-written and the discussions of interest. I recommend publication together with a revision of the points made below.
Requested changes
1 - When the authors say "there is a little mismatch between the dispersions close to the Dirac nodes", it would be nice to be more precise. Is the mismatch in the meV range? Is it in the 10meV range? Is it larger?
I think it would be best to show the band structure of the different models so that readers can jugde by themselves.
2- I am sorry I was not able to follow the sentence 'Since the tight-binding model cannot not reproduce exactly the energy dispersion obtained from the DFT calculations anyway, we ignore such terms from now on'. Why the TB model cannot not reproduce the DFT results?
3- I am sorry I was not able to certainly understand this sentence: "the leading k-independent spin-orbit coupling gives rise to the conservation of the spin projection on a particular axis near the Γ-point, which is consistent with the sample, edge- orientation, and gate-voltage independent spin quantization axis on edges of a sample". What is meant by "consistent with the sample, edge-orientation"? Does it means that it is at fixed angle away from the edge? Perhaps this sentence could be broken into peaces that makes its comprehension easier.
4- It could be useful to clarify why the Berry curvature appears in the field-induced Hall effect. Most readers will understand why it usually appears in the anomalous Hall effect (zero-field, by definition a different experiment).
5- I don't understand why smaller spin-orbit coupling leads to larger Hall effect in Eq. 25. At least in the conventional AHC, in linear response, the AHC vanishes without SOC.

---

## Round 3 · Referee Report · Anonymous (Referee 1) · 2022-2-22

Report

The authors have addressed my comments/questions and I suggest publication of the paper as it is.

---

## Round 3 · Author Response

We are grateful to the Referees for their comments and suggestions. We have uploaded our response to the referee comments in separate files.

---

## Round 3 · List of Changes

Summary of changes in response to referee report 1

1) We have included two new references (Refs.~31 and 32) in the revised version of the manuscript. (First Paragraph of Section-IIIC, Page 8)

2) In response to referee comment 2, we have added a clarifying sentence, stating that addition of sub-leading SOC would break spin conservation, but they don't. (To the end of second paragraph, left column , page 4)

3)We have modified the sentence in the revised manuscript in response to referee comment 3. (first paragraph, left column , page 5)

4)We have added a note on the magnitude of anomalous Hall effect to the ``Conclusions" section of the revised manuscript.

Summary of changes in response to referee report 2

1) We have mentioned the mismatch value in response to referee comment 1. (first paragraph, left column, page 3)

2)In response to referee comment 2, we have added explicit citations (refs. 27 and 28) to the text to emphasize the point as well as also added a follow-up sentence to clarify the point in question. (To the end of first paragraph, left column , page 3)

3) We broke the sentence into two parts in response to referee comment 3. (second paragraph, left column , page 4)

4) We have added a clarifying sentence right after Eq.(25) on page 8 in response to referee comment 4.

---

## Editorial Decision

published